# Polysaccharide Extracted from *Bletilla striata* Promotes Proliferation and Migration of Human Tenocytes

**DOI:** 10.3390/polym12112567

**Published:** 2020-11-01

**Authors:** Zhi-Yu Chen, Shih-Heng Chen, Chih-Hao Chen, Pang-Yun Chou, Chun-Chen Yang, Feng-Huei Lin

**Affiliations:** 1Institute of Biomedical Engineering, College of Medicine and College of Engineering, National Taiwan University, Taipei 100, Taiwan; d79010340217@gmail.com (Z.-Y.C.); shihheng@mac.com (S.-H.C.); 2Department of Plastic and Reconstructive Surgery, Chang Gung Memorial Hospital, Chang Gung University and Medical College, Taoyuan 333, Taiwan; chihhaochen5027@yahoo.com.tw (C.-H.C.); chou.asapulu@gmail.com (P.-Y.C.); 3Department of Materials Science and Engineering, National Taiwan University, Taipei 100, Taiwan; kh61604@hotmail.com; 4Institute of Biomedical Engineering and Nanomedicine, National Health Research Institutes, Miaoli 350, Taiwan

**Keywords:** tendon, polysaccharide, *Bletilla striata*, tenocytes, MEK/ERK1/2, PI3K/Akt, extracellular matrix

## Abstract

Tendon healing after injury is relatively slow, mainly because of the weak activity and metabolic properties of tendon cells (tenocytes). *Bletilla striata* polysaccharide (BSP) has been reported to enhance cell proliferation. Here, we aimed to increase tendon cell proliferation by BSP treatment. We isolated tenocytes from the flexor tendon of human origin. Moreover, we improved the process of extracting BSP. When human tenocytes (HTs) were treated with 100 μg/mL BSP, the MEK/ERK1/2 and PI3K/Akt signaling pathways were activated, thereby enhancing the proliferation ability of tenocytes. BSP treatment also increased the migration of HTs and their ability to secrete the extracellular matrix (Col-I and Col-III). In conclusion, BSP was successfully extracted from a natural Chinese herbal extract and was shown to enhance tenocytes proliferation, migration and collagen release ability. This study is the first to demonstrate improved healing of tendons using BSP.

## 1. Introduction

The tendon is a soft and dense tissue that mechanically links muscle and bone. It is made up of tenocytes (or tenoblasts) and extracellular matrix (ECM), which predominantly comprises type I collagen. The tendon functions as a buffer to prevent concentration of stress that would result from a direct muscle-to-bone connection [1,2,3,4]. Hand injuries account for up to 20% of all injuries requiring emergency treatment in the United Kingdom [5]. Tendon injuries are quite common and were reported in 54.8% of patients with a small laceration and 92.5% of patients with a deep injury in 2005 [6]. Each year, more than 30 million musculoskeletal injuries occur worldwide, of which over half are tendon and ligament injuries, and the actual number of injuries is likely to be even higher because many are not recorded [7,8,9].

Surgical treatment of tendon lacerations is currently the most common therapeutic approach and includes sutures (end to end sutures) and grafts [10]. Tendon healing can be divided into three overlapping stages: inflammation, regeneration (or proliferation), and remodeling. In the first stage (inflammation), the activated platelets release chemoattractants to let inflammatory cells, such as neutrophils and monocytes, migrate from circulation to the injury site. During the regeneration stage, tenocytes migrate to the repair site, proliferate, and begin to synthesize and secrete ECM mostly of type III collagen. In the final stage (remodeling), cell density and general synthetic activity are gradually decreased. the synthesis of type III collagen is replaced by the synthesis of type I collagen and ECM becomes more aligned. The remodeling phase begins 1–2 months after injury and can last more than 1–2 years [11,12]. Post-injury healing of tendons is typically poor due to the low inherent cellularity, low metabolic activity, and poor circulation of tendon tissues [9,13,14]. Therefore, increasing tenocyte proliferation and migration ability are key factors in improving tendon healing. For example, patellar tendon healing can be improved through enhancing the migration and proliferation of tenocytes via the bone morphogenetic protein 2 [15]. Platelet-rich plasma-clot release (PRCR) has been used to induce human tenocyte (HT) proliferation and collagen synthesis [16]; rat bone marrow mesenchymal stem cell-derived conditioned medium (MSC-CM) has also been shown to promote tenocyte proliferation and migration by activating the extracellular signal-regulated kinase1/2 (ERK1/2) signal molecules [17]. Although the use of growth factors, protein or PRP could effectively increase the proliferation of tenocytes, there are still concerns about high cost and easy degradation.

Herbal medicine, which is one of the main streams of traditional medicine, have been in use since the early days of mankind in around 75% of the world population. In the UK, Germany and France, in those developed countries, herbs or herbal extracts have seen a major increase in use and have even been used as prescription drugs [18,19]. In the last few years, there has been lots of research discussing the beneficial effects of herbs or herbal extracts on cell proliferation. In a 2014 review paper, it was mentioned that *Rhodiola*, *Ganoderma spore Polygala*, *Tetramethylpyrazine*, *Gardenia*, *Astragaloside* and *Ginsenoside Rg1* could promoted neural stem cells proliferation and *Acanthopanax*, *Angelica* could maintain neural stem cells survival [20]. Herbal extracts from *Persea americana*, flowers of *Althaea officinalis*, *Chamaemelum nobile*, *Thymus vulgaris*, leaves of *Rosmarinus officinalis* and *Urtica dioica* that exert positive effects on hair proliferation via ERK, Akt, cyclin D1, and Cdk4 signaling in dermal papilla cells (DPCs) [21]. *Bletilla striata* polysaccharide (BSP) is extracted from *B. striata (Thunb.)*, which is widely distributed in East Asian countries and has been used in traditional Chinese herbal medicine for treating bruises, alimentary canal mucosal damage, burns, bleeding, and ulcers [22,23]. BSP consists of (1→2)-α-d-mannopyranose and (1→4)-β-d-glucopyranose [24] and has been shown to have anti-inflammatory, anti-oxidation, and anti-tumor effects, as well as wound healing properties [25,26,27]. Recent studies have shown that BSP can also induce human umbilical vein endothelial cell (HUVEC) and human corneal endothelial cell (HCEC) proliferation [23]. Several previous studies have indicated that BSP is biocompatible and non-toxic and can be applied to many tissues such as the eyes, skin, and tumors. However, so far, there is no relevant literature on the application of BSP to tendons or tenocytes.

In this study, we first isolated tenocytes from the flexor tendon of the patients. We also developed a modified BSP extraction method, which is based on previously published work [28,29,30]. The effect of BSP application to tenocytes was evaluated based on cell proliferation, cell migration, and the ability of cells to secrete ECM. Finally, the signaling pathways involved in this process were explored.

## 2. Materials and Methods

### 2.1. Materials and Reagents

*Bletilla striata* (BS) was purchased from Sheng Chang Corporation (Taoyuan, Taiwan), and the WST-1 cell proliferation assay kit was obtained from Takara Bio USA, Inc. (Mountain View, CA, USA). Antibiotics, trypsin–EDTA, DMEM, MEM and FBS were purchased from GE Healthcare (Little Chalfont, UK). Live/dead kit, SuperScript first-strand synthesis system for RT-PCR, high-capacity cDNA reverse transcription kits, TaqMan real-time PCR master mixes, probes, and TRIzol were purchased from ThermoFisher Scientific (Waltham, MA, USA). All other chemicals were purchased from Sigma-Aldrich (St Louis, MO, USA).

### 2.2. Extraction of Polysaccharide from BS

The method of BSP extraction was modified from a previous study [28]. Dry BS powder was dispersed in 80 °C water for 4 h, and then centrifuged for 10 min at 5000× *g* to separate and collect the aqueous extract. The crude extracts were precipitated by an addition of 95% (*v/v*) ethanol and stored overnight at 4 °C. The resulting precipitate was collected by centrifugation at 5000× *g* for 10 min. The last two steps were repeated twice, and the precipitate was then resuspended in water. Deproteinization was performed by adding Sevage reagent (chloroform/n-butanol 4:1) to the solution and stirring overnight. The aqueous phase was collected by centrifugation, concentrated and dialyzed at a cut-off of 3500 Da (60035515, Orange Scientific, Braine-I’ Alleud, Belgium), and lyophilized to prepare the BSP. The structural characteristics of BSP were confirmed by FTIR spectral analysis, as well as ^13^C and ^1^H NMR spectral measurements. The molecular weight of the polysaccharide fractions was determined by GPC, and total sugar content was determined using a previously published phenol/sulfuric acid-based method [28].

### 2.3. Cytotoxicity of BSP

MTT assay: According to ISO 10,993 guidelines, the effect of BSP treatment on cell viability was evaluated by an MTT assay. L929 cells were seeded in a 96-well cell culture plate (10^4^ cells/well) in minimum essential medium (MEM, Sigma, M0643, UK) containing 10% FBS at 37 °C with 5% CO_2_ for 24 h. The culture medium was then replaced with 100 μL of positive control, negative control and different concentrations (1, 10, 100, 1000 μg/mL) of BSP for 24 h and 96 h. Subsequently, 10 μL of MTT solution was added to each well, and the cells were incubated at 37 °C for 4 h. The medium was substituted by 100 μL dimethylsulfoxide (DMSO) and the plates were shaken for 15 min, following which, absorbance was measured at 570 nm with an ELISA reader (MolecularDevices, SPECTRAmax Plus 384, San Jose, CA, USA). Positive control: zinc diethyldithiocarbamate (ZDEC; Sigma, 329703, St Louis, MO, USA). Negative control: aluminum oxide (Al_2_O_3_; Sigma, 11028, St Louis, MO, USA).

Live/dead staining: Live/dead viability/cytotoxicity assay kit was used according to the manufacturer’s instructions (Invitrogen, L3224, Waltham, MA, USA). L929 cells were cultured in MEM medium with BSP (0, 1, 10, 100, 1000 μg/mL) for 24 h, and then incubated with 2 μM Calcein AM and 4 μM ethidium homodimer-1 (EthD-1) for 30 min at room temperature in the dark. Cells were washed with MEM serum-free medium, and images were acquired using a fluorescence microscope. In this assay, green fluorescence (excitation/emission maxima 495 nm/515 nm) was considered indicative of live cells, and red fluorescence (excitation/emission maxima 528 nm/617 nm) was considered indicative of dead cells.

### 2.4. Isolation and Culture of HTs

Flexor tendon biopsies were obtained from four patients (age ranging from 18 to 55 years old). Tendon biopsies were approved by the Chang Gung medical foundation institutional review board (IRB) (Ref: 201601422B0). The method of HT isolation from the tendon was modified from a previously described method [16]. Tendon samples were cleaned of surrounding adipose tissues, cut into pieces, and digested in a solution of 0.5 U/mL collagenase (Serva, 17454, Heidelberg, Germany) in DMEM for 6 h. The resulting tenocyte/collagenase solution was collected and centrifuged at 200× *g* for 10 min room temperature. The supernatant was discarded, and the pelleted cells were gently washed in PBS to remove the residual collagenase. The isolated HTs were cultured in DMEM supplemented with 20% FBS, 1.5 mg/mL sodium bicarbonate, and 1% antibiotics (Antibiotic-Antimycotic (100X), Gibco, 15240-062, Waltham, MA, USA). The cells used in the experiment were all collected at the beginning of the second passage.

### 2.5. Gene Expression of HTs

HTs were seeded into 6-well cell culture plates (2 × 10^5^ cells/well) in DMEM and were incubated overnight. Total RNA of the HTs was extracted using Direct-zol™ RNA MiniPrep (Zymo Research, R2050, Irvine, CA, USA). First strand complementary DNA (cDNA) was synthesized using the SuperScript^®^ III first-strand synthesis system kit (Thermo Fisher Scientific, 18080-051, Waltham, MA, USA) according to the manufacturer’s instructions. Real-time PCR was performed on a StepOne real-time PCR system (Applied Biosystems) using TaqMan universal PCR master mix (2x) and specific primers for *Tenascin C* (*TNC*) [Hs01115665_m1], *Scleraxis* (*SCX*) [Hs03054634_g1], *Tenomodulin* (*TNMD*) [Hs00223332_m1], *Collagen type I* (*COL* I) [Hs00164004_m1] and *GAPDH* [Hs02786624_g1].

### 2.6. Western Blot Analysis

HTs were treated with BSP for various time periods then added with a lysis buffer containing a protease inhibitor and a phosphatase inhibitor at the end of the experiment. Lysates were centrifuged at 10,000× *g* for 10 min at 4 °C, and the supernatant was then collected. Equal amounts of protein from the cell lysates were resuspended in sample buffer and loaded onto a 10% SDS–PAGE gel for gel electrophoresis, followed by transferring the separated proteins onto a PVDF membrane and subsequently blocking with 5% (*w/v*) non-fat dried milk in TBST overnight at 4 °C. The membrane was incubated with the appropriate primary antibodies specific for Erk 1/2, p-Erk 1/2, Akt, p-Akt, CoL-I, CoL-III, and β-actin overnight at 4 °C; then, the membranes were incubated with the respective secondary peroxidase-conjugated antibodies. The proteins on the PVDF membrane were detected with the ECL detection system (Pierce) according to the manufacturer’s protocol.

### 2.7. Gap Closure Migration Assay

HTs were seeded into a 24-well plate. A line was scratched into the cell layer using a 200 µL tip, followed by three washes with PBS [31,32]. Fresh medium containing 100 μg/mL of BSP in DMEM was added to the cells. The average scraped area in each well was measured, and the change in area for each experimental condition was compared with that of the control. Migration rate was normalized by comparison with the initial gap distance in each group.

### 2.8. Transwell Migration Assay

The upper transwell chambers (Corning Costar, Cambridge, MA, USA) were loaded with 10^5^ HTs in 200 μL of DMEM containing 2% FBS, and the lower chambers with 500 μL of DMEM containing 20% FBS and 100 μg/mL of BSP. Following incubation for 12 h, cells in the upper chamber were removed, and the membranes were fixed in 4% paraformaldehyde for 20 min. The cells that had migrated to the lower side of the filter were then stained with 0.1% crystal violet for 10 min and observed under a light microscope. Crystal violet was dissolved in 300 μL of 33% acetic acid, and the absorbance was measured at 573 nm with an ELISA reader (MolecularDevices, SPECTRAmax Plus 384, USA).

### 2.9. ECM Synthesis Test

HTs were seeded into 6-well cell culture plates (2 × 10^5^ cells/well) in DMEM and were incubated overnight. HTs were treated with 100 μg/mL of BSP for 48 h then analyzed in culture medium and cells simultaneously. The culture medium were collected then analyzed the total collagen secretion according to the manufacturer’s instructions by hydroxyproline assay kit (Biovision, K226-100, Milpitas, CA, USA). HTs were added with a lysis buffer containing a protease inhibitor and phosphatase inhibitor than analyzed the protein expression levels of CoL-I and CoL-III according to the 2.6 Western Blot procedure.

### 2.10. Statistical Analysis

All data are expressed as mean ± standard deviation (SD) from three to six independent experiments. Statistical differences between groups were tested by Student’s *t*-test or one-way analysis of variance (ANOVA) post hoc tests and by Tukey’s test using GraphPad Prism software (GraphPad Software, Inc., La Jolla, CA, USA), and a probability (*p*) value of less than 0.05 (*p* < 0.05) was considered statistically significant.

## 3. Results

### 3.1. Characterization and Cytotoxicity of BSP

We optimized the polysaccharide precipitation step of the BSP extraction process. For the precipitation of BSP, four sample volumes of 95% (*v/v*) ethanol were found to be more effective than three sample volumes used in previous studies [22,28,30]. Additionally, we extended the incubation period from 3–4 h at room temperature (RT) to overnight at 4 °C. For improved extraction, we repeated the precipitation and incubation steps. Using the modified BSP extraction scheme, 23.3 ± 0.1 g (Table 1) of polysaccharide was obtained from 100 g of dry B. striata. The yield increased from the original 17.1% to 23.3% and was significantly higher than that obtained from the original method (*p* < 0.01). There was no significant difference in the total sugar content compared to that in previous studies. BSP molecular weight (Mw) has been reported to be 135 kDa or 190 kDa, whereas the molecular weight of the BSP extracted in this study was 165 kDa by gel permeation chromatography (GPC). Protein and nucleic acid were not detected using UV-vis absorption spectroscopy measured in the range of 200–400 nm.

Fourier-transform infrared spectroscopy (FT-IR) fingerprinting of BSP was performed to identify distinct peaks in the range of 500–4000 cm^−1^. To demonstrate good reproducibility, the FT-IR spectra of BSP obtained from two different extraction experiments using the optimized method are shown in Figure 1. The identical bands in the two spectra indicate the stability of the product in addition to demonstrating the reproducibility of the method. The absorption peaks at 3343, 2910, 1650, and 1382 cm^−1^ are typical for polysaccharides, peaks at 1023 cm^−1^ and 1150 cm^−1^ represent pyranoses, whereas the peaks at 809 cm^−1^ and 880 cm^−1^ represent mannose and β-glucosyl residues, respectively. C−O−C glycosidic and C−O−H side group band vibrations were observed in the spectra from 1000 to 1200 cm^−1^. All absorption bands seen in this study (Figure 1A) were consistent with those obtained in previous studies [33,34]. The ^1^H (4.55 ppm and 4.80 ppm) and ^13^C (103.50 ppm and 101.11 ppm) NMR spectrum of BSP showed two main peaks, which were predicted to represent β-glucopyranose and α-mannopyranose repeating units in BSP; these were not significantly different compared with those reported in the previous literature. The cytotoxicity of different concentrations of BSP (1–1000 μg/mL) on L929 cells was examined by MTT assay (Figure 2A). The viability of all BSP-treated groups (24 and 96 h) was higher than 70%. In accordance with ISO 10993-5, BSP was determined to have no cytotoxicity potential. The live/dead staining (Figure 2B) results showed that L929 cells had good viability after treatment with different concentrations of BSP. In the BSP cytotoxicity test with L929 cells, we also observed that the cell activity of L929 cells were significantly increased (*p* < 0.05) in 100 and 1000 μg/mL of BSP two groups on day 1 (24 h), even on day 3 (96 h), the cell activity of L929 treated with 100 μg/mL BSP group was still significantly increased (*p* < 0.05).

### 3.2. Identification of HTs

Tenomodulin (TNMD), a late marker of the mature tenogenic phenotype; scleraxis (SCX), an important tenogenesis transcription factor; and tenascin C (TNC), an early marker of embryonic tendons, are common tendon markers specific to the tenogenic phenotype [35,36]. To further distinguish HTs and human scar fibroblasts (HSF), RT-PCR was used to identify the specific markers of tenocytes after isolation of the cells (Figure 3). Since HSF also secretes collagen type I, Figure 3A(a) shows that there is no significant difference in collagen type I gene expression between HT and HSF. However, as shown in Figrue 3A(b–d), scleraxis (SCX), Tenomodulin (TNMD) and Tenascin C (TNC), specific gene expression was significantly higher (*p* < 0.05) in HTs than in HSFs. Figure 3B shows the immunocytochemistry (ICC) staining for TNMD (OriGene, TA338487). Two images in the Figure 3B left column are the ICC staining results of HTs and HSFs under 40× magnification, and in the right column are the results of 100× magnification. As indicated by the green fluorescence, TNMD protein expression was higher in HTs than in HSFs. The results of TNMD ICC in the HTs positive staining, once again verified the result of *TNMD* between HTs and HSFs in the RT-PCR experiment. Specific gene RT-PCR and ICC staining experiments confirmed that the cells isolated in this article are HT instead of HSF.

### 3.3. HT Proliferation Test

HT cell viability at different concentrations of BSP (1–1000 μg/mL) was examined using a WST-1 assay (Figure 4A). When HTs were treated with BSP, their activity was promoted, with the most remarkable effect obtained at a concentration of 100 μg/mL (*p* < 0.01). Therefore, for subsequent experiments, we maintained a BSP concentration of 100 μg/mL in the experimental group. As shown in Figure 4B(a), it was observed that when HTs were treated with BSP for 2 days, the number of cells was significantly higher than in the control group as counted using the handheld automated cell counter (Scepter™ 2.0 Cell Counter, MERCK Millipore, USA). It was also possible to observe the proliferation of HTs after adding BSP in the photomicrograph, as shown in Figure 4B(b). The above results indicate that BSP does increase the HT proliferation ability.

Activation of MEK/ERK1/2 and PI3K/Akt signaling pathways is associated with cellular proliferation, migration, differentiation and cell survival [37,38,39,40,41]. To investigate whether BSP activates the MEK/ERK1/2 and PI3K/Akt signaling pathway, HT cells were treated with BSP100 for various time periods and the expression of ERK1/2, phosphorylated-ERK1/2 (p-ERK1/2), Akt and phosphorylated-Akt (p-Akt) was analyzed using Western blotting. As illustrated in Figure 5A, after adding BSP to HTs, significant phosphorylation of ERK1/2 and Akt was observed at 6 h (*p* < 0.05) and was greater after 24 h—specifically, 27.6 and 22.4 times greater than ERK1/2 and Akt phosphorylation at 0 h group, respectively (*p* < 0.001). Quantification by Western blotting (Figure 5B) showed that BSP activates ERK and Akt in HTs in a time-dependent manner.

To confirm the pathway based on Figure 5, we selected PD98059 (PD), one of the MEK/ERK1/2 pathway inhibitors, and wortmannin (Wort), a PI3K/Akt pathway inhibitor, for the cell pathway inhibition experiment [42,43,44]. HTs were treated with two different concentrations of PD98059 (10 and 50 μM) and wortmannin (0.5 and 1 μM) for 1 h and stimulated with 100 μg/mL BSP for 24 h. As shown in Figure 6, PD98059 significantly inhibited the MEK/ERK1/2 pathway at 10 μM (*p* < 0.01) with maximum inhibition at 50 μM (Figure 6A,B). Wortmannin inhibited the BSP-induced phosphorylation of Akt at 0.5 μM (*p* < 0.001). The maximum inhibitory concentration of wortmannin was 1 μM with no significant difference from that of the 0.5 μM concentration (Figure 6C,D). We thus selected 50 μM PD98059 and 1 μM wortmannin as inhibitors for HTs treated with BSP, then counted the cell number. Results are shown in Figure 6E; PD98059 inhibited BSP-induced proliferation of HTs by 47% compared with BSP group and wortmannin inhibited it by 60%, while co-treatment of both inhibitors inhibited BSP-induced proliferation of HTs by 32%.

### 3.4. HT Migration Test

Gap closure migration experiment results (Figure 7A,B) showed that when the time of HT treated with BSP was increased to 24, 48, 72, and 96 h, the BSP100 group HT showed significant differences in the gap remaining (*p* < 0.05) compared with the control HT group. Although HTs cell morphology observed in Figure 7A are all spindle-shaped, in BSP 48h and 96h groups, the migratory HTs are more elongated than the control cell morphology. As shown in Figure 7B, when HT was added with BSP for 96 h, cells migrated throughout the scratch area; however, the control HT group still showed a 38% gap area.

The gap closure method is only a preliminary observation of migration, which may be caused by cell proliferation. Therefore, we used the transwell method to further confirm the BSP-induced migration of HTs. As illustrated in Figure 7C, crystal violet staining indicated that after 12 h, more cells in the BSP group were attracted to the lower layer than in the control group. The OD value was measured by dissolving crystal violet in acetic acid to obtain the results shown in Figure 7D. The OD value results showed that the migration ability of the BSP group was significantly different from that of the control group (*p* < 0.05).

### 3.5. HT ECM Synthesis Test

A hydroxyproline assay kit (Biovison, K226-100, USA) was then used to quantify the total collagen content in the culture medium. It was calculated assuming that 12.5% of collagen is hydroxyproline [36,45]. Based on the results shown in Figure 8A, the secretion of total collagen in the HT group to which BSP was added was significantly higher (*p* < 0.05) than in the control group. Finally, collagen type I and collagen type III were further analyzed by Western blotting. Results displayed in Figure 8B,C show that when BSP is added to HT, the secretion of type I and type III collagens is significantly increased, and the secretion of type III collagen is increased more significantly.

## 4. Discussion

In traditional Chinese herbal medicine, *Bletilla striata* (BS) is considered to be efficacious against hemostasis and inflammation and to promote tissue regeneration and bodily functions [46]. However, most of the previous studies on BSP have focused on its anti-oxidation and anti-inflammatory effects, while only a few studies have investigated its other effects. Only in recent years have there been relevant studies on its effects in promoting tissue healing (such as wound healing) and cell proliferation. As BS was previously used on wounds to promote rapid healing, we were interested in elucidating whether BSP can increase tissue repair and cell proliferation. This study is the first to treat HTs with BSP and examine the consequent effects on cell proliferation, cell migration, and the ability to secrete ECM.

Accumulating evidence has revealed that BSP can induce cell proliferation in different types of cells such as HUVEC, HCEC, L929 and 3T3 [29] and even HTs (in this research). However, the underlaying molecular mechanisms remain to be fully elucidated. This article uses a series of pathway activation and inhibition experiments to explore how BSP-induced HT cells to proliferate. Particularly, when BSP is added to HTs, the levels of phosphorylated ERK1/2 and Akt gradually increased. There remains a significant difference in intensity even at 48 h, which is only slightly decreased compared to that at 24 h. Previous studies have shown that phosphorylation of ERK1/2 or Akt induced by drugs or substances will reach a maximum level over a period of 6–12 h, and then gradually decrease [41,47]. When kiwi fruit polysaccharides were added to normal human dermal fibroblasts (NHDF) and culture for more than three days, cell proliferation results were significantly different from those of the control group [48]. MCF-7 cells, breast cancer cell lines, treated with *Lycium barbarum* polysaccharide (LBP) showed that the phosphorylated ERK expression continued to increase to 24 h [49]. We inferred that polysaccharides are slowly degraded in the culture medium, thereby continuously inducing ERK1/2 or Akt phosphorylation.

Cell migration is fundamental for proper immune response, wound healing, and tissue homeostasis to establish and maintain the organization. The locomotory cycle of cell migration consists of cells protruding and adhering at the leading margin, developing contractile forces between the front and trailing margins, and finally releasing trailing adhesions. Cells first adhere at some point and cytoplasmic displacement at leading edge are common features of cell migration [50,51]. We could observed from the image result of gap closure cell migration experiment in this article that the morphology of HT treated with BSP has the above-mentioned features of cell migration. Although this study did not use more than 40x magnification microscope to observe, the migratory HTs are obviously elongated and spindle-shaped. As mentioned above, it has been suggested that rat bone marrow MSC-CM promotes tenocyte proliferation and migration by activating the ERK1/2 pathway and increasing cytoskeletal polymerization in cellular and nuclear stiffness [17]. The results of this study showed that BSP-induced HT proliferation not only activates the MEK/ERK1/2 pathway, but also activates the PI3K/Akt pathway. Choroidal endothelial cell (CEC) proliferation was induced through the activated MEK/MAPK and PI3K/Akt pathways and PI3K cross-talk with ERK1/2 [52]. ERK1/2 plays an important role in cell migration. A review paper demonstrated the effect of ERK1/2 on cell migration through its interaction with different proteins (such as Vinexin β and Calpain 2) [53]. In 2018, a report confirmed via pharmacologically inhibiting the ERK1/2 pathway that endostatin can increase myofibroblast migration through activation of the pathway [44]. In this study, we inferred that BSP could induce HT migration through activation of the ERK1/2 pathway.

Recently, the application of BSP combined with a carrier in animal experiments for wound healing has shown promising results [29,54]. BSP in vitro and in vivo studies have shown a possible wound healing function of BSP by regulating the activity of macrophages during the early stages and either increasing or decreasing inflammatory response to accelerate cell proliferation. BSP could simultaneously accelerate fibroblast (or myofibroblast) proliferation and collagen depositions during the cell proliferation period. Finally, BSP continues to induce collagen synthesis until granulation tissue formation and epithelial cell thickening are achieved [46].

Although we did not discuss macrophages in their early stages, they can still be observed, since HT can induce cell proliferation upon BSP treatment and increase the synthesis of ECM type I and type III collagens, similar to the cell proliferation phase of the process proposed in the review paper above [46]. We tried to evaluate molecular pathways in this study to explore BSP-induced HT cell proliferation and migration. The result shows that activation of the MAPK/ERK1/2 and PI3K/AKT pathways is associated with HT cell proliferation, but the underlaying molecular mechanism remains unclear. In the future, we will further investigate which receptor on HT BSP binds to in order to influence the activation of the MAPK/ERK1/2 and PI3K/AKT pathways. Additionally, we will also explore the relationship between ERK1/2 pathway activation and HT cell migration in more detail (such as the expression of the MMP family).

## 5. Conclusions

The results of this study show that the improved BSP extraction method described here can be used for consistent and stable extraction of BSP from BS, with a higher yield than that obtained using the conventional extraction method. From a series of cell proliferation experiments, it could be seen that BSP at an optimal concentration of 100 μg/mL promotes HT proliferation by approximately 1.2-fold via activating the Erk and Akt pathways. BSP induces HT migration as indicated by a gap closure and a transwell migration assay. Moreover, BSP increases HT collagen I and III secretion. The results of this study show that BSP, obtained from a Chinese herbal extract, has potential for enhancing the proliferation and migration of HTs. Further research is necessary to test various combinations of carrier and BSP, in order to improve tendon healing.

## Figures and Tables

**Figure 1 polymers-12-02567-f001:**
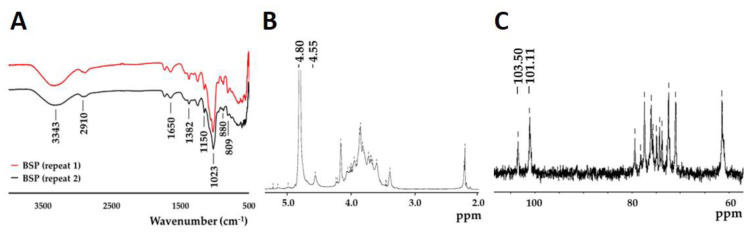
Characterization of *Bletilla striata* polysaccharide (BSP). (**A**) FT-IR spectrum; (**B**) ^1^H-NMR spectrum; (**C**) ^13^C-NMR spectrum.

**Figure 2 polymers-12-02567-f002:**
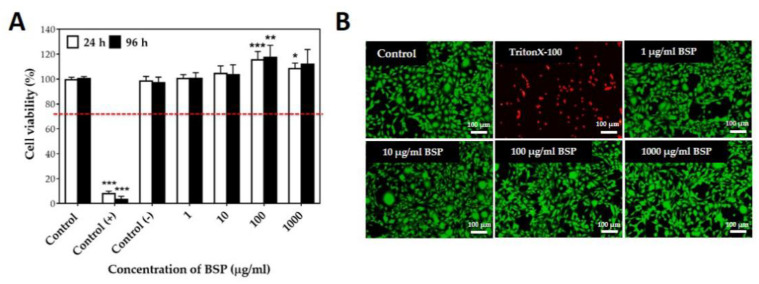
Cytotoxicity of *Bletilla striata* polysaccharide (BSP) based on L929 cells treated with BSP. (**A**) MTT assay. Control (+): ZDEC (zinc diethyldithiocarbamate); Control (−): Al2O3. * *p* < 0.05, ** *p* < 0.01, *** *p* < 0.001 compared with control group, *n* = 6. (**B**) Live/dead staining (24 h).

**Figure 3 polymers-12-02567-f003:**
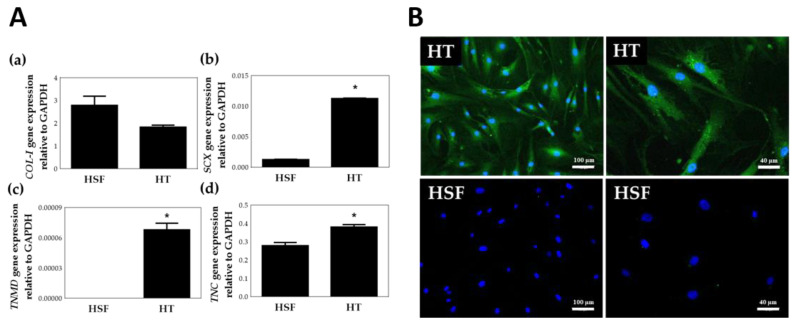
Identification of human tenocytes (HTs). (**A**) Gene expression of HTs and human scar fibroblasts (HSFs) by TaqMan RT-PCR. (**a**) *collagen type I* (*COL-I*), (**b**) *scleraxis* (*SCX*), (**c**) *Tenomodulin* (*TNMD*) and (**d**) *Tenascin C* (*TNC*). * *p* < 0.05 compared with HSF group, *n* = 6 (**B**) TNMD Immunocytochemistry staining of HTs (40×, 100×) and HSFs (40×, 100×). Green: Tenomodulin (TNMD), Blue: Hoechst 33342.

**Figure 4 polymers-12-02567-f004:**
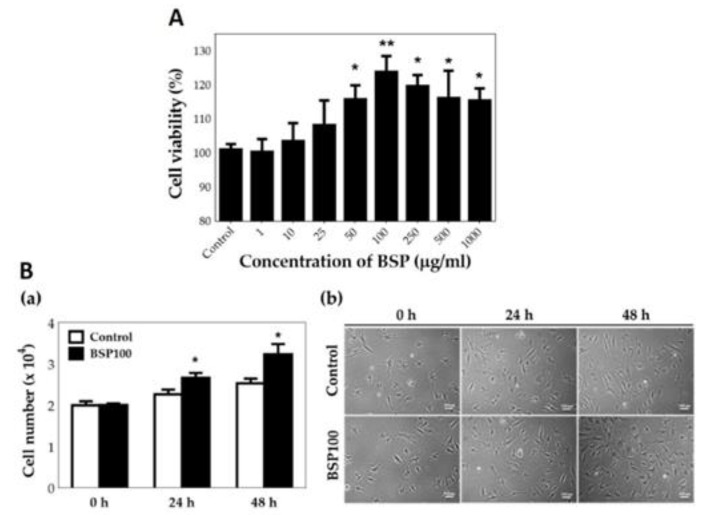
(**A**) Cell viability of human tenocytes (HTs) treated with *Bletilla striata* polysaccharide (BSP) for 48 h examined by WST-1 assay. (**B**) Cell number of HTs treated with BSP for 0 h, 24 h, 48 h examined by (**a**) handheld automated cell counter, (**b**) photomicrograph. BSP100: 100 μg/mL of BSP, * *p* < 0.05, ** *p* < 0.01 compared with each control group, *n* = 6.

**Figure 5 polymers-12-02567-f005:**
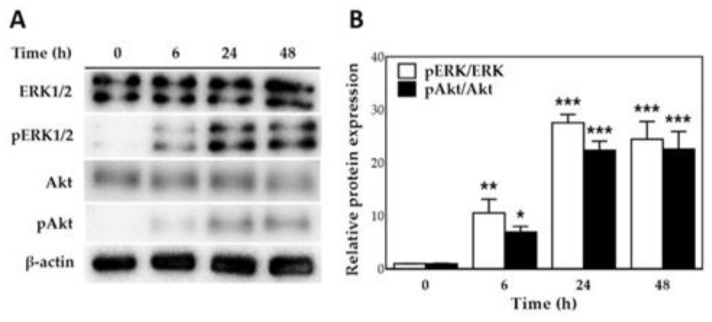
*Bletilla striata* polysaccharide (BSP) stimulated phosphorylation of ERK and Akt in human tenocytes (HTs). (**A**) Western blot analysis of HTs treated with 100 μg/mL of BSP for various time. (**B**) Quantification of Western blot. * *p* < 0.05, ** *p* < 0.01, *** *p* < 0.001 compared with (0 h) group, *n* = 3.

**Figure 6 polymers-12-02567-f006:**
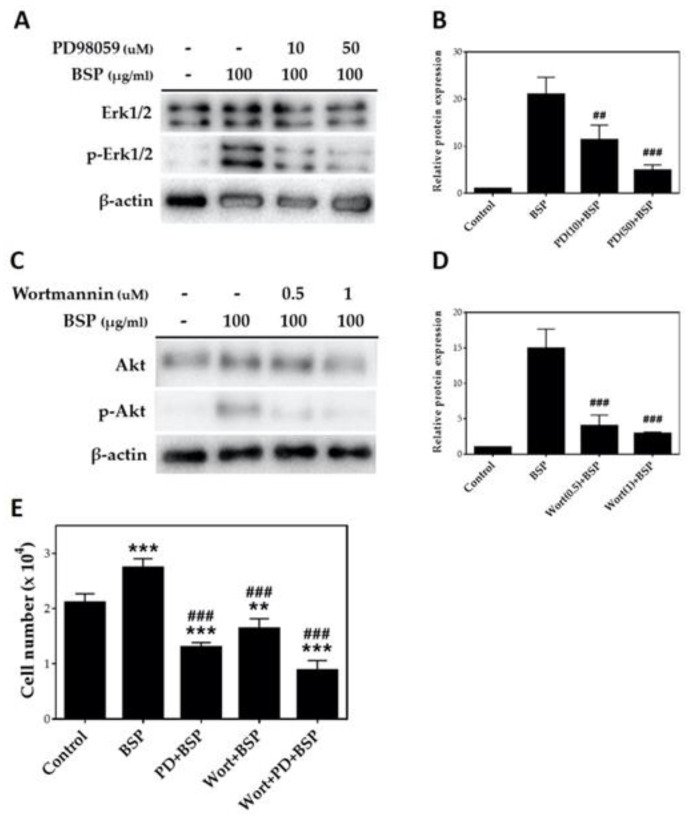
Inhibition experiment of *Bletilla striata* polysaccharide (BSP) induced phosphorylation of ERK1/2 and Akt by (**A**,**B**) MEK/ERK1/2 inhibitor and (**C**,**D**) PI3K/Akt inhibitor were examined using Western blotting. (**E**) Cell number of human tenocytes (HTs) treated with BSP in the presence or absence of PD98059 and wortmannin pre-treatment. ** *p* < 0.01, *** *p* < 0.001 compared with the control group. ## *p* < 0.01, ### *p* < 0.001 compared with BSP group, *n* = 3.

**Figure 7 polymers-12-02567-f007:**
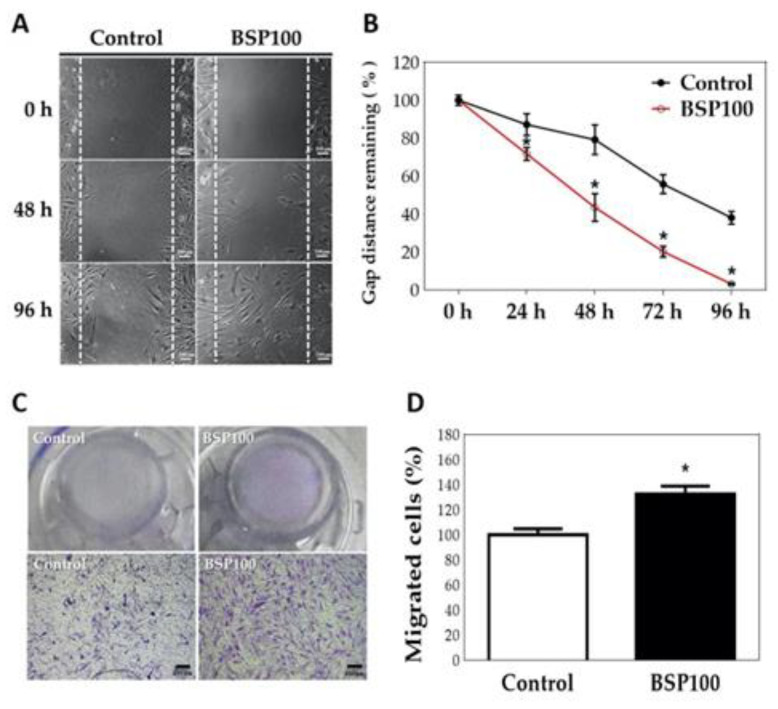
Gap closure migration assay of human tenocytes (HTs) treated with *Bletilla striata* polysaccharide (BSP). (**A**) Photomicrograph. (**B**) Quantification of gap distance. Transwell migration of HTs treated with BSP for 12 h. (**C**) Morphology. (**D**) Quantification of Western blot. * *p* < 0.05 compared with control group, *n* = 3.

**Figure 8 polymers-12-02567-f008:**
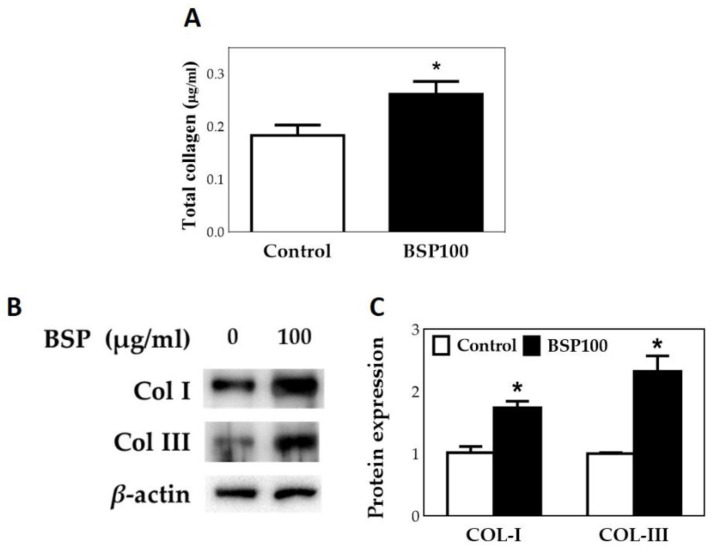
Human tenocytes (HTs) treated with *Bletilla striata* polysaccharide (BSP) for 48 h. (**A**) Total collagen assay. (**B**) Western blotting for CoL I and CoL III. (**C**) Quantification of Western blot. * *p* < 0.05 compared with control group, *n* = 3.

**Table 1 polymers-12-02567-t001:** Comparison of yield, total sugar content, and molecular weight between previous and modified BSP extraction methods.

	Yield (%)	Total Sugar Content (%)	Molecular Weight (kDA)
Previous BSP extraction method	17.1 ± 0.2	68.5 ± 4.3	198
Modified BSP extraction method	* 23.3 ± 0.1	73.9 ± 1.9	165

* *p* < 0.05, *n* = 3.

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
