# Peer review of "Polysaccharide Extracted from Bletilla striata Promotes Proliferation and Migration of Human Tenocytes"

_polymers, 2020, doi:10.3390/polym12112567_

Round 1

Reviewer 1 Report

Polysaccharide Extracted from Bletilla striata 3 Promotes Proliferation and Migration of Human 4 Tenocytes

  1. Introduction needs to be modified.
  2. Fig 3 B: Labelling was not explained properly in the legend as well as in the text.

As mentioned by authors, TNMD protein expression was not different in both HTs and HSFs. Further, for both cells, scaling was different. Authors need to present them in the same scale and also quantification of the fluorescence intensity needs to be performed to distinguish the differences mentioned by the authors.

  1. Did the authors analysed the activation of ERK, Akt pathways earlier than 6 hours?

        To further conclude that the BSP activates ERK and Akt in a time            dependent manner, as the authors claim, authors also need to present the western blotting of different time points between 30 min and 6 hours.

  1. Are the ERK and Akt results represented by Western blots in Fig.4A from the same blot or from different blots?
  2. Fig 8: Uniformity should be maintained in explaining the results. The authors showed the increase in Col-I at the gene and protein level which was missing in case of Col-III.
  3. Repetition of explaining the results was observed in discussion which should be avoided.
  4. Discussion should be improved.

Reviewer 2 Report

The manuscript entitled “Polysaccharide extracted from Bletilla striata promotes proliferation and migration of human tenocytes” written by Chen et al. is about studying the effect exhibited by Bletilla striata polysaccharide (BSP) in several in vitro experiments including cell proliferation and collagen secretion. This topic is timely, fits in completely with the aims of the journal, and might have an important contribution to tissue engineering applications. The manuscript is well-written, quite interesting, and preliminary data can be convincing. However, I find several issues that prevent me to endorse its acceptance at the present stage.

Minor comments

Abstract; Line 26. The authors stated that “BSP was successfully extracted from a natural Chinese herbal extract and was shown to enhance tendon healing”. In my opinion, the present manuscript has only collected data from in vitro cultures of human tenocytes. I think that one should have more evidence in vivo to be sure that the present strategy is able to promote tendon healing.

Page 2. Line 88. According to the authors, BSP was confirmed by FT-IR, 1H-NMR, and 13C-NMR. However, a FT-IR spectrum is only provided on Figure 1. The authors should also include the corresponding 1H and 13C NMR spectra in the same figure.

Page 9. Line 286. HT ECM Synthesis Test. This protocol should be included in the Materials and Methods section.

Major comments

Page 2. Line 92. Incubation times should be enlarged up to 96 hours in order to study the long-term toxicity effect of BSP when combining with L929 cells.

Figure 8. I acknowledge the effort carried out by the authors to quantify the total amount of collagen in the culture medium. However, it would be more convenient to enlarge the incubation times up to 96 hours at least to measure the BSP long-term effect on total collagen synthesis. In addition, the authors should use quantitative immunofluorescence analysis to detect collagen I in the cytoplasm.  

Reviewer 3 Report

I found the manuscript written by Chen et al. on “Polysaccharide Extracted from Bletilla striata Promotes Proliferation and Migration of Human Tenocytes” interesting. They showed Bletilla striata polysaccharide (BSP) enhances tenocytes proliferation in vitro and studied MEK/ERK1/2 and PI3K/Akt signaling pathways supporting tenocytes ability to proliferate.

In general, the manuscript is not well written and requires thorough revision and proofreading, with special attention needed for sentence fluency and clarity. The text is often awkwardly organized and unnecessarily wordy. Care must be taken that some sections of the paper do not become run-on descriptions of previous sections with insufficient synthesis for the reader to draw meaningful conclusions including abstract and conclusion.

From the scientific point of view:

  1. I highly encourage the authors to include morphological assessment of human tenocytes in addition to proliferation and migration as part of their study given how critical the phenotypic appearance of cells is when it comes to the regulation of cell activities.
  2. Addition of NMR data on BSP can provide more descriptive information on characterization of BSP and help the reader better follow the context.
  3. Does BSP have any absorbance measure around 500-600 nm? If so, was that subtracted from all raw values for MTT analysis?
  4. Although identification of human tenocytes (HT) from HSF is an interesting measure to validate these cells, but it is not clear why authors did not look into up/down regulation of any of these cytoskeleton markers (COL-I, scleraxis, Tenomodulin, and Tenascin C) when it comes to treatment with BSP and how BSP can influence their remodeling activity at gene expression levels.
  5. It is not very clear why WST-1 assay was chosen to assess proliferation given both MTT (used in Figure 1A) and WST-1 assays are based on tetrazolium salts converted by dehydrogenases to the corresponding formazan, which are widely used to assess cell viability and proliferation, however it is not clear to draw a same conclusion from described findings in Figure 1A vs. Figure 4A when it comes to dose dependency.
  6. For HT ECM synthesis evaluation, it is crucial to look at collagen gene expression after addition of BSP more than just one concentration to see if there is a correlation between the secretion of total collagen and the addition of BSP and if so, what would be the minimum effective concentration compared to the control group
  7. Given the presented data by the authors, it is so unclear and insufficient to draw a conclusion on the effect of BSP on HTs using MEK/MAPK 331 and PI3K/Akt pathways. Detailed study design is highly needed to show the effect of BSP and including additional positive and negative controls (inhibitors) to block these pathways and show their relationship with BSP when it comes activation/inactivation in order to draw a meaningful conclusion.

Round 2

Reviewer 1 Report

The authors responded accordingly.

Reviewer 2 Report

The authors have successfully addressed the reviewer's comments. I recommend this manuscript for publication in Polymers

Reviewer 3 Report

I appreciate the authors' efforts to adequately addressing each of my concerns and providing feedback for each one. I personally feel comfortable to recommend the work for publication after minor English editing and grammatical formatting.